# Vegetable Response to Added Nitrogen and Phosphorus Using Machine Learning Decryption and the N/P Ratio

**Léon Etienne Parent** 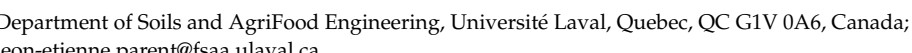

Department of Soils and AgriFood Engineering, Université Laval, Quebec, QC G1V 0A6, Canada; leon-etienne.parent@fsaa.ulaval.ca

**Abstract:** The current N and P fertilization practices for vegetable crops grown in organic soils are inaccurate and and may potentially damage the environment. New fertilization models are needed. Machine learning (ML) methods can combine numerous features to predict crop response to N and P fertilization. Our objective was to evaluate machine learning predictions for marketable yields, N and P offtakes, and the N/P ratio of vegetable crops. We assembled 157 multi-environmental fertilizer trials on lettuce (*Lactuca sativa*), celery (*Apium graveolens*), onion (*Allium cepa*), and potato (*Solanum tuberosum*) and documented 22 easy-to-collect soil, managerial, and meteorological features. The random forest models returned moderate to substantial strength ($R^2$ = 0.73–0.80). Soil and managerial features were the most important. There was no response to added P and null to moderate response to added N in independent universality tests. The N and P offtakes were most impacted by P-related features, indicating N–P interactions. The N/P mass ratios of harvested products were generally lower than 10, suggesting P excess that would trigger plant N acquisition and possibly alter soil N and C cycles through microbial processes. Crop response prediction by ML models and ex post N/P ratio diagnosis and N and P offtakes proved to be useful tools to guide N and P management decisions in organic soils.

**Keywords:** muck vegetables; yield-limiting factors; random forest; universality tests; N and P excess; Redfield ratio; N and P stoichiometry

## 1. Introduction

Current nitrogen (N) and phosphorus (P) fertilizer recommendation models rely primarily on soil and tissue test calibration based on a limited number of multi-environment fertilizer trials (MEFTs) and on field surveys [1–3]. Questionable nutrient budget models based on nutrient offtakes and efficiency were thought to support recommendations led to insurance applications, overfertilization, and economic loss [4]. Extra N dosage is often applied by growers against uncertainty [5], resulting in economic loss, environmental damage [6,7], altered crop quality [8], and greater crop susceptibility to pest attacks [9]. On the other hand, the insurance P fertilization of vegetable crops contributed to P loss and the eutrophication of surface waters [10]. Traditional methods to make fertilizer recommendations be revisited in relation to system's sustainability.

The MEFTs are conducted under the ceteris paribus assumption [11]. Such assumption fails at the step of assembling results from several MEFTs because climatic, managerial, and soil factors vary widely and simultaneously among the experimental sites [12]. Using large and diversified datasets, machine learning (ML) models can combine relevant site-specific features to predict yield response to added nutrients [13]. Although complex models such as ML models may show high accuracy, universality tests are still required to verify the model's generalization capacity to real cases in growers' fields unseen by the model [14,15].

On the other hand, the N and P cycles are stoichemically related by N/P ratios specific to life systems to reach stable protein/RNA ratios [16–19]. At vegetation level in short-term fertilization trials, N/P mass ratios less than 10 correspond to N-limited production, and

N/P mass ratios exceeding 20 indicate P-limited biomass production [17]. However, plants and the soil microbial biomass must compete for soil available N and P forms. The well-constrained atomic C:N:P ratio in the microbial biomass (60:7:1) may become a useful tool to assess nutrient limitation to plants in terrestrial ecosystems [18,20]. In high-P soils, N could become limiting to both microbes and plants despite the high N mineralization rates often reported in organic soils [21].

The N/P ratio is generally interpreted phenomenologically as N or P limitation. In natural peat soils, sub-optimal N supply is controlled by P availability [22]. However, too low a N/P ratio depends on whether the value at numerator (N) is too low or the value at denominator (P) is too high. The nutrient limitation hypothesis does not hold in intensive vegetable production where extra N and P fertilization rates are applied due to high crop value and thus N and P are not limiting crop yield. Hence, the N/P ratio should be interpreted conversely as relative excess rather than relative limitation.

Because the N/P stoichiometry must be maintained in plants [17] and the soil microbial biomass that scavenges for soil C, N, and P, and strives for N/P homeostasis [18,20], new questions are raised:

1. Is there higher N fertilizer requirement where plant P nutrition is managed at excessive levels? This could occur in high-P soils;
2. Is there higher P fertilizer requirement where plant N nutrition is managed at excessive levels? This could occur in high-N mineralizing soils.

Growers can improve N and P fertilization decisions using ex ante ML-assisted site-specific crop response to added N and P, ex post N and P offtakes, and a wise interpretation of the N/P mass ratio of the final product. We hypothesized that ML models are accurate to predict marketable yields, N and P offtakes, and the N/P ratio in the harvested product. Our objective was to apply data science to fertilization trials to avoid growers' inclination toward extra N and P rates that endanger the sustainability of on- and off-farm agroecosystems. The database of N and P fertilizer trials on vegetables grown in organic soils comprised several managerial, soil, and meteorological features, marketable yields, and N and P offtakes.

## 2. Materials and Methods

### 2.1. Data Source

The database included 79 nitrogen (N) and 75 phosphorus (P) fertilizer trials, as well as three potassium (K) fertilizer trials, conducted by several research teams on vegetable crops grown in organic soils south of Montreal, Quebec, Canada (45.015 to 45.277 latitude north, and −73.707 to −73.374 longitude west) during the 1995 to 2008 period (Table 1). Such soils, also called Histosols or peat soils, contain ≥30% organic matter [23]. Where nutrient rates and timings were varied at an experimental site, other nutrients were applied following state recommendations. Fertilizer treatments were arranged as a randomized complete block design with three to four replications per site.

Target variables were marketable yields, N and P offtakes, and the N/P mass ratio between N and P offtakes. Marketable yields were measured in two central rows. Five plants were harvested randomly in each plot, oven-dried at 65 °C for 24 h to 36 h, ground to pass through a 1-mm sieve, composited, and analyzed for total P by plasma emission spectroscopy (ICP-OES) after tissue digestion [24]. Total N was quantified by Dumas combustion (Leco-2000 instrument, St Louis, MO, USA). Crop N and P offtakes were assessed as nutrient concentration in the dry matter (65 °C) times the dry biomass of the harvested product.

Key relevant yield-impacting features are listed in Table 2. Soil analyses were conducted for samples collected in the 0–20 cm layer at springtime before crop establishment. Soil acidity was reported as pH in water. Soil organic matter was quantified by Dumas combustion (LECO 2000 equipment, St Louis, MO, USA) as C and N concentrations. In older studies where soil organic matter was quantified by ashing or using the Walkey–Black procedure and for N by micro-Kjeldahl, conversion to total C and N was obtained after

proper calibration. The C and N concentrations reflect the potential rate of N mineralization in organic soils [21]. Phosphorus and metals were extracted using the Mehlich3 method [25] then quantified by plasma emission spectroscopy. Soil P saturation was computed as the $[P/(Al + 5Fe)]_{Mehlich3}$ molar ratio [10].

**Table 1.** Fertilizer trials conducted on vegetables grown in organic soils of southwestern Quebec, Canada.

| Common Name | Latin Name | Trials | | |
|---|---|---|---|---|
| | | **N** | **P** | **K** |
| | | **Number of Sites** | | |
| Head lettuce | | 10 | 12 | 0 |
| Leaf lettuce | *Lactuca sativa* | 7 | 3 | 0 |
| Romaine lettuce | | 8 | 7 | 0 |
| Celery | *Apium graveolens* | 23 | 21 | 3 |
| Onion | *Allium cepa* | 18 | 17 | 0 |
| Potato | *Solanum tuberosum* | 13 | 15 | 0 |
| | | 79 | 75 | 3 |

**Table 2.** Lists of features to be related to target variables.

| Feature | Unit | Range |
|---|---|---|
| 1.   $pH_{water}$ | pH unit | 4.31–7.65 |
| 2.   Total C | % on weight basis | 9.7–54.3 |
| 3.   Total N | % on weight basis | 0.61–4.04 |
| 4.   P soil test | $mg\ P\ kg^{-1}$ | 10–1337 |
| 5.   K soil test | $mg\ K\ kg^{-1}$ | 64–2189 |
| 6.   Ca soil test | $mg\ Ca\ kg^{-1}$ | 4087–22,183 |
| 7.   Mg soil test | $mg\ Mg\ kg^{-1}$ | 378–3810 |
| 8.   Cu soil test | $mg\ Cu\ kg^{-1}$ | 0.9–65.1 |
| 9.   Zn soil test | $mg\ Zn\ kg^{-1}$ | 5.6–51.1 |
| 10.   Mn soil test | $mg\ Mn\ kg^{-1}$ | 7–164 |
| 11.   Fe soil test | $mg\ Fe\ kg^{-1}$ | 202–1739 |
| 12.   Al soil test | $mg\ Al\ kg^{-1}$ | 0–1632 |
| 13.   N dosage | $kg\ N\ ha^{-1}$ | 0–300 |
| 14.   Number of split N applications | - | 0 to 3 |
| 15.   P dosage | $kg\ P_2O_5\ ha^{-1}$ | 0–240 |
| 16.   K dosage | $kg\ K_2O\ ha^{-1}$ | 0–375 |
| 17.   Seeding/planting date | Julian day | 121–210 |
| 18.   Harvest date | Julian day | 174–287 |
| 19.   Seasonal precipitations (seeding/plantation to harvest) | Cumulated rainfall (mm) | 45–469 |
| 20.   Irrigation | Yes/No | Yes/No |
| 21.   Shannon diversity index for rainfall (seeding/plantation to harvest) | - | 0.44–0.78 |
| 22.   Seasonal heat units (seeding/plantation to harvest) | Degree-days $\geq 5\ °C$ | 434–1883 |

Seeding and plantation dates were obtained from growers. Previous vegetable crops in the rotations were assumed to contribute negligibly to N supply in the following year. Lettuce and celery crops were irrigated following local models. Irrigation water was supplied to 72.5% of the onion plots. Potato crops were not irrigated. Meteorological data were retrieved from the closest meteorological stations using the geographic coordinates and the year of experimentation. The Shannon diversity index (*SDI*) that reflects rainfall distribution during the growing season was computed as follows [26]:

$$SDI = \left[ -\sum_{i=1}^{n} \frac{p_i ln(p_i)}{ln(n)} \right]$$

where $p_i$ is the fraction of daily rainfall relative to seasonal precipitation ($n$ days); there is complete evenness of daily rainfall where $SDI = 1$; where $SDI = 0$, all rain falls within a single day (complete unevenness).

### 2.2. Machine Learning Model

Random forest (RF) generates decision trees from the random extraction of features. This bagging method averages predictions from sampling with replacement. The RF was run using the Orange Data Mining freeware v. 3.34.0 programmed in the Python language (University of Ljubljana, Ljubljana, Slovenia). Decision-tree models such as RF separate subsets recursively about cutoff values in the training dataset to minimize the variance of the target variable until a prescribed minimum number of instances is reached. The number of trees was tuned at 50. No split subsets were smaller than five instances. The model was run using random sampling stratified by category (crop type and site number as categorical variables) rather than across observations. Indeed, model overfitting occurs where observations from the same trial (site) are assigned to both the training and testing subsets.

A subset made of one trial per crop picked at random was set apart to assess model's ability to generalize to unseen cases. The remaining database was partitioned between the training subset (75%) to build the model, and the testing subset (25%) to measure model accuracy or strength. Because the size of the N and P offtake and N/P databases were only 2/3 that of the marketable yield database, universality tests to test model's capacity to generalize were run for marketable yields only. General statistics are presented for N and P offtakes and the N/P mass ratio.

The relative importance of features was ordered using the RReliefF algorithm based on the nearest neighbor paradigm after considering feature interactions [27]. Model accuracy was reported as $R^2$ coefficient, $RMSE$ (root mean square error), and $MAE$ (mean absolute error), as follows:

$$R^2 = 1 - \frac{\sum_{i=1}^{n}(y_i - \hat{y}_i)^2}{\sum_{i=1}^{n}(y_i - \hat{y}_i)^2}$$

$$RMSE = \sqrt{\frac{1}{n}\sum_{i=1}^{n}(y_i - \hat{y}_i)^2}$$

$$MAE = \frac{1}{n}\sum_{i=1}^{n}|y_i - \hat{y}_i|^2$$

where $y_i$ is the observed target variable, $\hat{y}_i$ is the predicted target variable, $\hat{y}_i$ is the mean of observed target variables, and $n$ is the number of observations. The coefficient of determination ($R^2$) for the relationship between predicted and observed yields was interpreted as model strength as follows [28]: $R^2 < 0.25$, very weak; $0.25 \leq R^2 < 0.50$, weak; $0.50 \leq R^2 < 0.75$, moderate; $R^2 \geq 0.75$: substantial.

## 3. Results

### 3.1. Model Accuracy

The RF model relating marketable yields to 22 managerial, soil, and meteorological features returned substantial strength with $R^2 = 0.805$ (Table 3). Nonetheless, yield predictions varied widely about the mean as shown by $RMSE$ and $MAE$ (Figure 1). Indeed, not all yield-impacting features were documented by research teams due to the different objectives and financial constraints. The RF models for the N/P ratio and N offtake were less accurate than that for P offtake. The offtake and N/P models will need to be reinforced with additional field trials and the contribution of growers to universality tests.

**Table 3.** Model $R^2$ coefficient, *RMSE* (root mean square error), and *MAE* (mean absolute error) relating crop N and P offtakes and N/P ratios to features.

| Target | Features | RMSE | MAE | $R^2$ |
|:---:|:---|:---:|:---:|:---:|
| | | Mg ha$^{-1}$ | | |
| Yield | Crop, pH, soil C, soil N, soil test P, K, Ca, Mg, Cu, Zn, Mn, Fe, Al, N-P-K fertilization, number of split N applications, seeding/plantation date, harvest date, precipitations, heat units, SDI | 8.2 | 5.6 | 0.805 |
| | | kg ha$^{-1}$ | | |
| N offtake | Crop, pH, soil C, soil N, soil test P, N-P-K fertilization, number of split N applications, seeding/plantation date, harvest date, precipitations, heat units, SDI | 26.9 | 18.7 | 0.732 |
| P offtake | Crop, pH, soil C, soil N, soil test P, N-P-K fertilization, number of split N applications, seeding/plantation date, harvest date, precipitations, heat units, SDI | 5.4 | 3.8 | 0.800 |
| Crop N/P mass ratio at harvest | Crop, pH, soil C, soil N, soil test P, N-P-K fertilization, number of split N applications, seeding/plantation date, harvest date, precipitations, heat units, SDI | 0.81 | 0.54 | 0.743 |

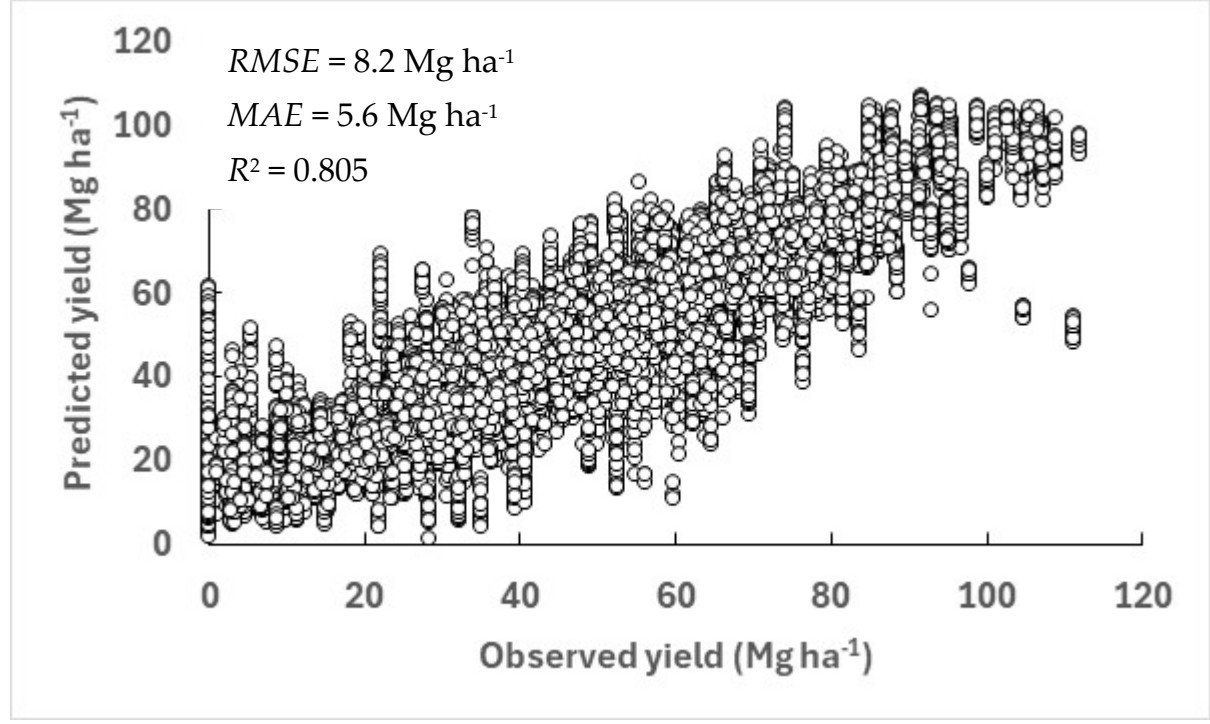

**Figure 1.** Relationship between predicted and measured marketable yields of vegetables grown in organic soils using 22 features. *RMSE* = root mean square of error; *MAE* = mean absolute error; $R^2$ = coefficient of determination.

The RReliefF algorithm ranked features in order of importance for marketable yields (Figure 2). Nutrient management features such as number of N splits, N and P dosage, crop type, and seeding/plantation and harvest dates, as well as soil tests appeared to be important. Meteorological features were of lesser importance. Because all features, but irrigation that was confounded with crop type, contributed to some degree to crop yield and were easy to collect, they were included in the ML model to make yield predictions.

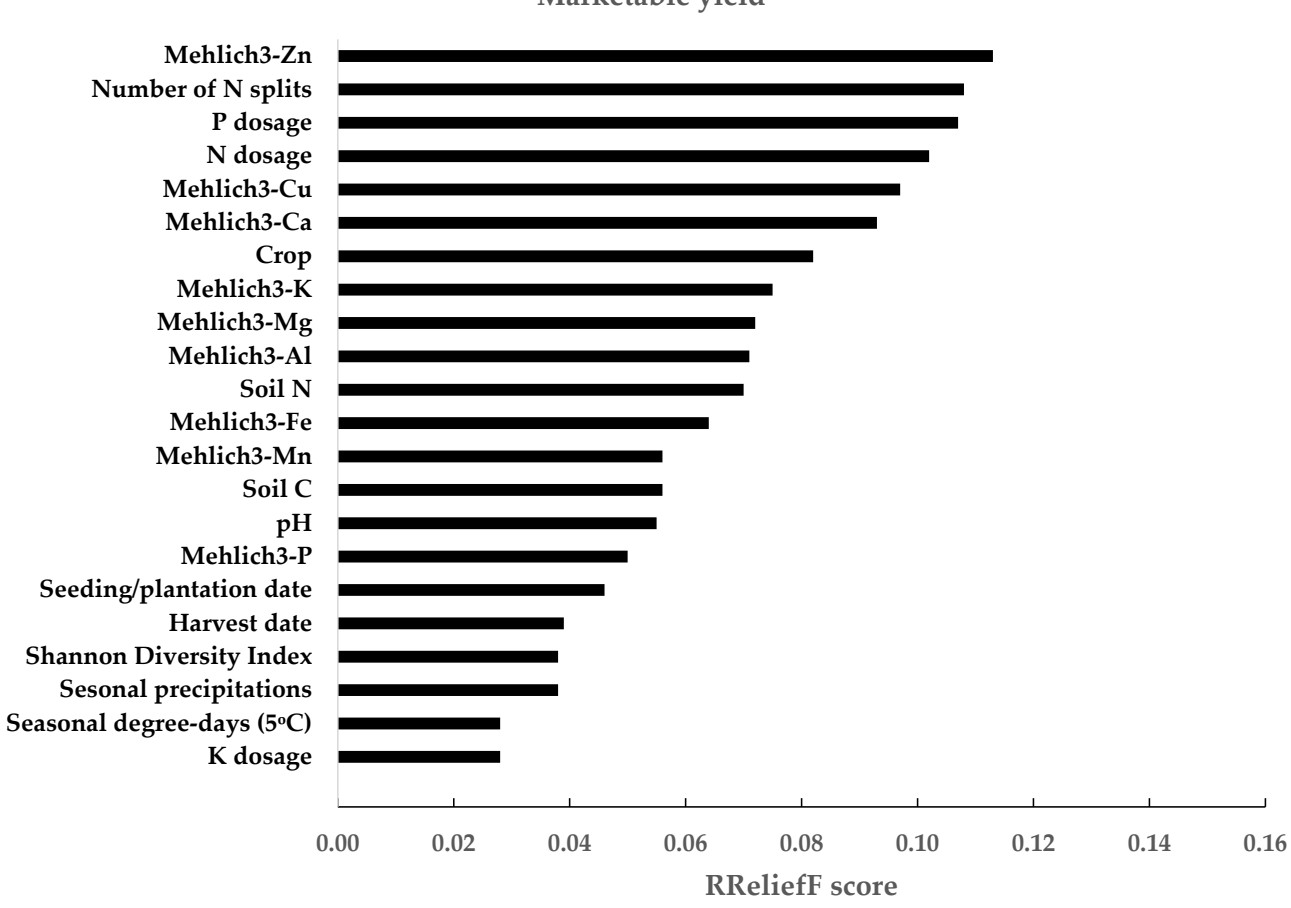

**Figure 2.** RReliefF scores of feature importance for yields of vegetables in organic soils.

Marketable yield, N offtake, and P offtake varied widely among crops (Table 4). While the N/P mass ratios varied between 1.2 and 11.6 and overlapped across crops, 99.5% of the observations showed N/P mass ratios < 10, indicating relative N limitation or P excess. Median ratios varied little from 3.3 to 4.8 among crops. The RReliefF scores indicated that crop type, P-related features, and soil N were important features in relation to N and P offtakes and the N/P mass ratio (Figure 3), confirming the central role of P features in the N and P cycles in those vegetable agroecosystems.

**Table 4.** Statistics on fresh yield, N and P offtakes, and the N/P ratio of the harvested portion of the crop.

| Crop | Statistics | Marketable Yield | N Offtake | P Offtake | N/P Ratio |
|---|---|---|---|---|---|
| | | Mg ha$^{-1}$ | kg ha$^{-1}$ | | |
| Celery | Minimum | 2.7 | 4.3 | 1.5 | 1.2 |
| | Median | 44.7 | 70.9 | 19.9 | 3.3 |
| | Maximum | 87.4 | 146.5 | 40.0 | 9.6 |
| Lettuce | Minimum | 0.9 | 8.0 | 1.8 | 2.4 |
| | Median | 26.3 | 57.2 | 11.4 | 4.8 |
| | Maximum | 77.0 | 258.5 | 53.1 | 9.9 |
| Onion | Minimum | 6.4 | 16.2 | 3.0 | 2.1 |
| | Median | 50.2 | 74.2 | 21.8 | 4.7 |
| | Maximum | 66.3 | 175.6 | 42.3 | 10.3 |
| Potato | Minimum | 14.4 | 37.8 | 8.9 | 1.6 |
| | Median | 35.6 | 96.5 | 20.0 | 4.5 |
| | Maximum | 58.0 | 282.2 | 69.6 | 11.6 |

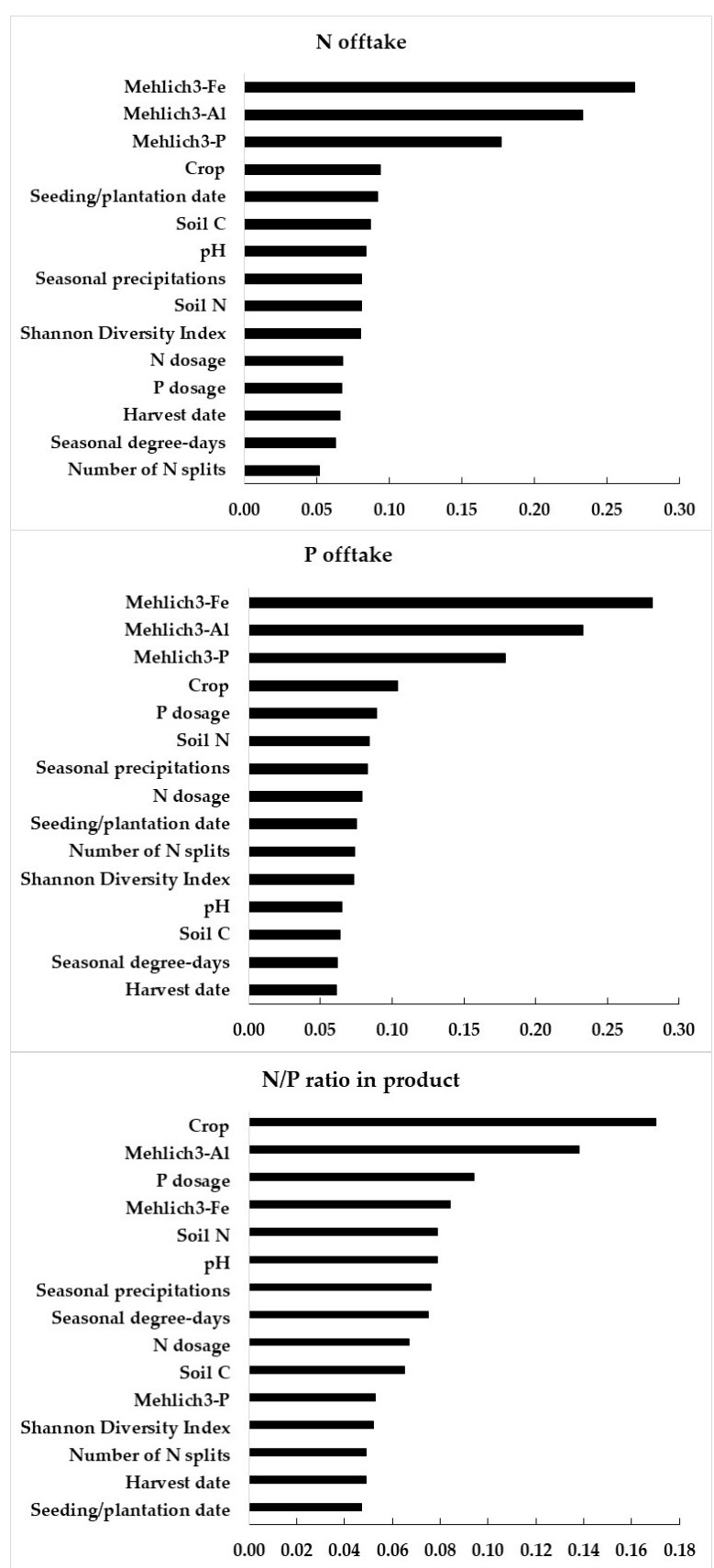

**Figure 3.** The RReliefF scores order the relative importance of features impacting crop N and P removal and the N/P ratio in the harvested portion of the plant.

### 3.2. Universality Tests to Predict Marketable Yields

Universality tests were run on one field per crop by combining the 22 features to predict yield response to N and P additions (Table 5). Features were within the ranges reported in Table 2. Soil pH, the C/N ratio, soil P saturation index, and Mehlich3-extractable elements varied widely among growers' sites. Crop response trajectories were obtained by varying the N and P rates at each site.

**Table 5.** Features used to run eight universality tests in growers' fields. The C/N ratio and P saturation index were added to facilitate interpretation.

| | | Site Number | | | | | | | |
| --- | --- | --- | --- | --- | --- | --- | --- | --- | --- |
| | | N Tests | | | | P Tests | | | |
| | **Feature (Unit)** | **1** | **2** | **3** | **4** | **5** | **6** | **7** | **8** |
| 1. | Crop | Head lettuce | Celery | Onion | Potato | Head lettuce | Celery | Onion | Potato |
| 2. | $pH_{water}$ | 6.11 | 6.09 | 5.32 | 5.13 | 5.74 | 6.29 | 6.34 | 5.50 |
| 3. | Total C (%) | 49.2 | 38.3 | 44.3 | 42.3 | 48.4 | 40.3 | 33.9 | 44.2 |
| 4. | Total N (%) | 1.90 | 1.99 | 1.50 | 2.09 | 2.01 | 2.04 | 1.71 | 1.97 |
| 5. | Mehlich3 P soil test (mg kg$^{-1}$) | 159 | 261 | 148 | 498 | 101 | 223 | 608 | 199 |
| 6. | Mehlich3 K soil test (mg kg$^{-1}$) | 463 | 260 | 438 | 706 | 586 | 241 | 376 | 402 |
| 7. | Mehlich3 Ca soil test (mg kg$^{-1}$) | 11,200 | 14,780 | 9195 | 13,770 | 12,333 | 13,303 | 8336 | 15,538 |
| 8. | Mehlich3 Mg soil test (mg kg$^{-1}$) | 858 | 1792 | 3302 | 1239 | 1934 | 1756 | 1202 | 1818 |
| 9. | Mehlich3 Cu soil test (mg kg$^{-1}$) | 31 | 10 | 6 | 15 | 28 | 10 | 14 | 12 |
| 10. | Mehlich3 Zn soil test (mg kg$^{-1}$) | 32 | 24 | 24 | 15 | 21 | 24 | 14 | 10 |
| 11. | Mehlich3 Mn soil test (mg kg$^{-1}$) | 39 | 26 | 31 | 88 | 58 | 23 | 43 | 28 |
| 12. | Mehlich3 Fe | 425 | 781 | 357 | 899 | 625 | 688 | 670 | 846 |
| 13. | Mehlich3 Al soil test (mg kg$^{-1}$) | 3 | 63 | 343 | 256 | 8 | 68 | 434 | 147 |
| 14. | N dosage (kg ha$^{-1}$) | 0–120 | 0–210 | 0–150 | 0–150 | 90 | 140 | 120 | 100 |
| 15. | Number of split N applications | 1–2 | 1–3 | 1 | 1 | 1 | 1 | 2 | 1 |
| 16. | P dosage (kg ha$^{-1}$) | 60 | 100 | 80 | 90 | 0–120 | 0–160 | 0–120 | 0–200 |
| 17. | K dosage (kg ha$^{-1}$) | 120 | 200 | 150 | 100 | 120 | 200 | 150 | 180 |
| 18. | Seeding/planting date (Julian day) | 143 | 153 | 129 | 172 | 160 | 153 | 134 | 151 |
| 19. | Harvest date (Julian day) | 201 | 243 | 247 | 276 | 213 | 243 | 231 | 262 |
| 20. | Seasonal precipitations from seeding/plantation to harvest (mm) | 2.18 | 290 | 377 | 3.29 | 220 | 290 | 373 | 3.59 |
| 21. | Shannon diversity index for rainfall from seeding/plantation to harvest | 0.70 | 0.68 | 0.73 | 0.74 | 0.68 | 0.68 | 0.75 | 0.73 |
| 22. | Seasonal heat units (seeding/plantation to harvest) (degree-day) | 903 | 1469 | 1630 | 1463 | 874 | 1469 | 1543 | 1594 |
| | C/N ratio | 26.0 | 19.2 | 29.3 | 20.2 | 24.0 | 19.7 | 19.9 | 22.5 |
| | Mehlich3 P saturation index: molar ratio as $100 \times [P/(Al + 5Fe)]_{Mehlich3}$ (%) | 13.8 | 11.7 | 10.8 | 17.6 | 5.9 | 11.3 | 25.9 | 8.0 |

In general, yield response to added N was null in prediction (Figure 4), indicating the great N supply capacity of the tested organic soils as shown by C/N ratios less than 30. However, observed yields appeared to level off at 50 kg N ha$^{-1}$ for potato and at 30 kg N ha$^{-1}$ for lettuce. Measured onion yields were higher than predicted ones and seemed to level off at 100 kg N ha$^{-1}$, indicating more favorable growing conditions at the tested site compared to sites showing similar features in the database. However, there was short-distance variability as shown by differences in onion yields between replications. Note that the number of the N trials was not so large. More trials are required.

Observed and predicted crop responses to added P were negligible (Figure 5) due to high soil test P levels and soil P saturation ratios across sites. Potato, a high-P demanding crop, showed no response to added P even though the soil showed the lowest soil test P and P saturation index (Table 3). For other crops, where soil test P and the P saturation index were higher, the P fertilization was also ineffective. Hence, there was great potential to reduce the environmental footprint of P fertilization of those vegetable crops. Again, large yield variation

at plot scale, especially in onion trials, indicated that factors not documented in the database, such as those causing short-distance soil variability, could have impacted crop yield. Because the database was relatively small, more trials and universality tests are needed.

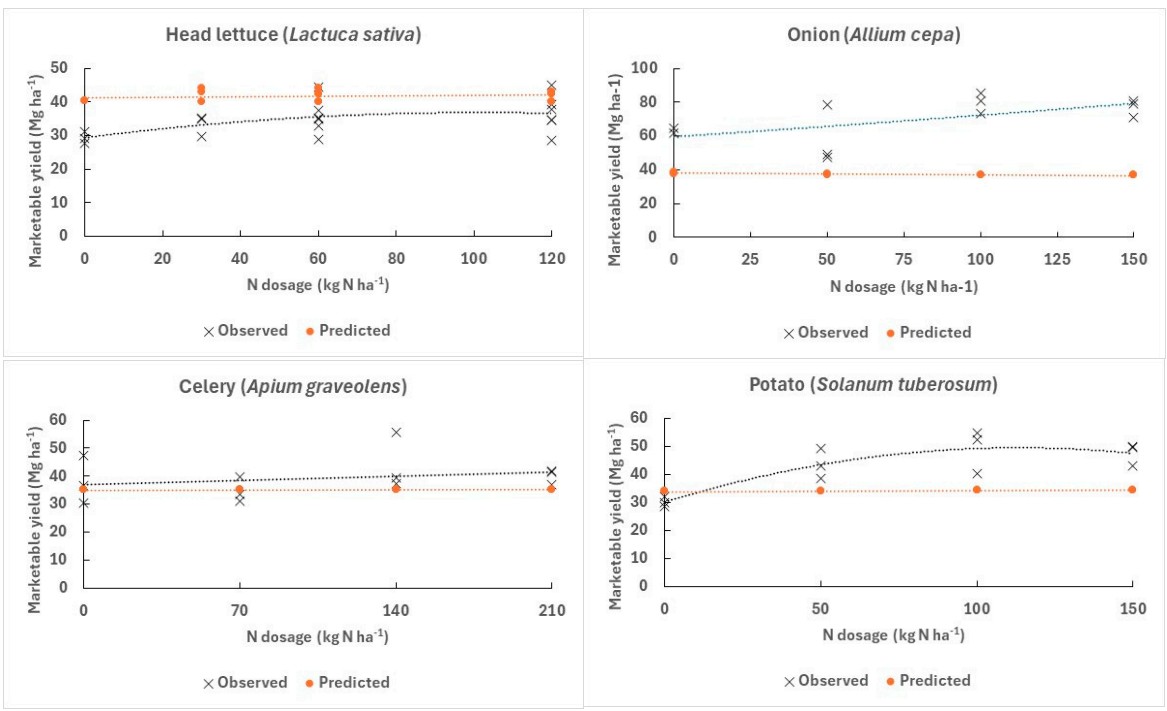

**Figure 4.** Patterns of crop responses to added N in organic soils. Predicted yields show no response. Observed yields show some response. Black dashed lines indicate plateauing. Quadratic trends for head lettuce and potato and linear trend for onion are misleading due to plateauing. Celery shows one outlier at 140 kg N ha$^{-1}$ and may require additional tests.

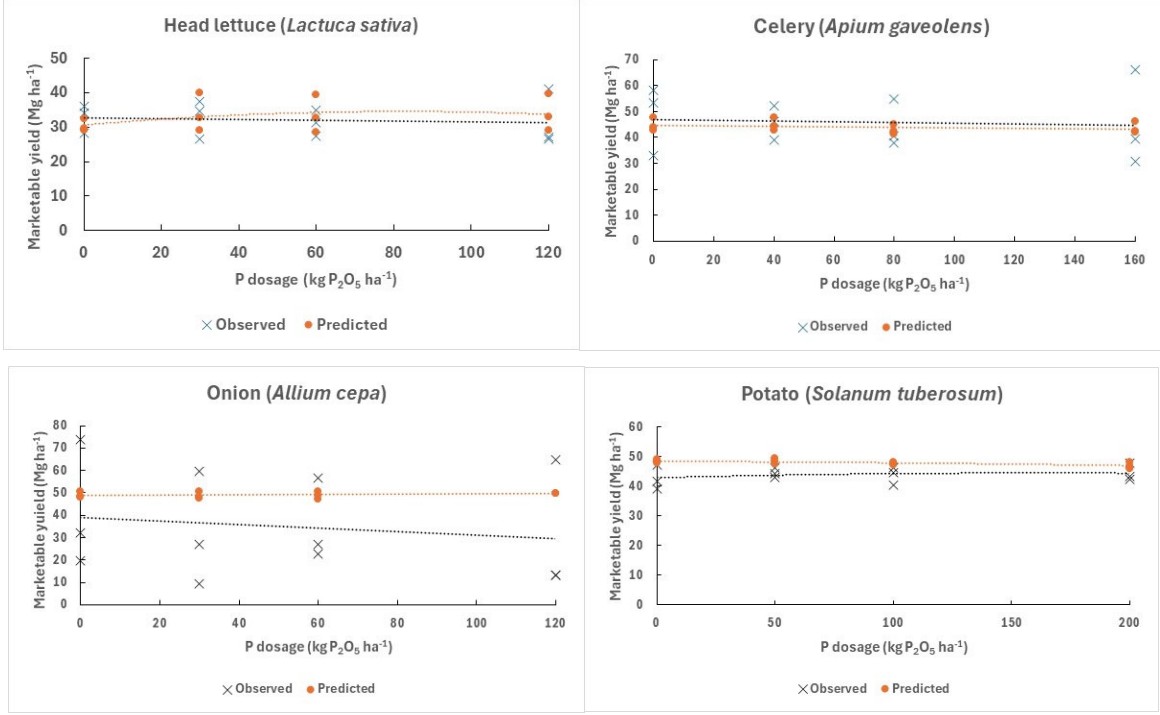

**Figure 5.** Patterns of crop responses to added P in organic soils. Predicted and observed yields show no well-defined response to added P.

## 4. Discussion

### 4.1. Model Accuracy

Using 22 features, the RF model to predict marketable yields from added N and P in organic soils showed substantial strength ($R^2$ = 0.805) across four vegetable crops. Indeed, features not documented in the database could contribute to model variation. In comparison, the accuracy of ML models for multi-environmental trials reached 0.80 for maize (*Zea mays*) [13] in Quebec, Canada. Given the substantial strength of the RF model to predict the marketable yields of vegetables, optimum growing conditions could be defined for a given species. Due to data dispersion (Figure 1) that may affect predicted yield and optimum N and P dosage at field scale, universality tests are still needed to compare model outputs to observations made in grower's fields, and contribute increasing the size and diversity of the vegetable database.

### 4.2. Universality Tests for N

Universality tests showed null to moderate responses of predicted and observed yields to added N due primarily to high N mineralization rates [21]. However, the right N rate to apply was difficult to assess because yields either responded linearly and quadratically at low N rates or tended to plateau. Selecting other response curves to determine optimum dosage is challenging because functions showing similar $R^2$ values may return contrasting optimum rates [29,30].

Eliminating N could be risky in organic soils for early-planted crops despite large amounts of organic N mineralized later in the season [31]. The decision to apply N would thus depend on planting date and seasonal climate. Due to the importance of N rate and timing as shown by RReliefF scores, the N fertilization could also be split into several applications adjusted to soil N supply during the season. Indeed, an alternative model could be built and updated using features and the climatic conditions prevailing before split N application [13].

### 4.3. Universality Tests for P

Universality tests for P indicated no response to added P above the P saturation index of 5%, supporting earlier results using traditional soil calibration methods [10]. Hence, there was little risk to reduce or eliminate P fertilization. In a high-P organic soil south of Lake Ontario, there were no adverse consequences where P was omitted during three consecutive years [31]. Hence, there is a great potential to abate the P load without yield loss in organic soils showing a P saturation index more than 5%, leading to economic and environmental gains as supported by universality tests in grower's fields.

### 4.4. The Need to Document More Yield-Limiting Factors

The N delivery and crop N demand are difficult to synchronize in space and time due to yearly variations in soil N supply from soil organic matter and crop residues, and in the nitrification, nitrate leaching, and denitrification processes [8]. Soil N mineralization potential depends on drainage, soil properties such as organic matter quality during organic soil transformations, crop management, temperature, and rainfall, especially early season rainfall, and the spatial distribution of soil water content.

A single harvest was established on experimental sites each year. The main crop may also be combined with another crop established earlier in the season that leaves crop and fertilizer residues on the soil after harvest. The main crop could also be followed by a cover crop to mop up easily accessible nutrient residues and make them available to the next-year crop. Such practices could be tested in the future.

Soil classification and drainage were not documented in the database. Organic soils are highly productive but vary widely in composition and evolution [32,33]. Organic soils of southwestern Quebec, reclaimed for agriculture in the early 20th century, are made of shallow shore swamps and deeper basin bogs deposited in channels and depressions [34]. Following the drainage of the original peatland, reclamation, and intensive cultivation,

organic soils undergo profound physical, chemical, and biological transformations [35]. During soil alteration, illuviated humus accumulates at the bottom of the arable layer, forming a layer of low permeability [36]. Such layer may become brittle upon irreversible drying. Prismatic and platy structure may even form in upper layers [37] and accelerate the leaching of nitrate, phosphate, and soluble carbon.

Peat thickness less than 60 cm shows signs of soil degradation [38]. Soil thickness and quality may vary widely [39]. Peat thickness and the presence of the humus illuviation layer could be documented in future studies. On the other hand, crops can be affected locally by other phenomena such as wind erosion, excess water, splash and rill erosion, surface sealing, and pest attacks. Precision agriculture could delineate soil management zones to account for limiting factors [39,40].

Soil nutrient limitations for vegetables grown on organic soils could also be assessed using tissue tests [3,31]. Nutrient ranges have been suggested for several vegetables grown in Quebec, Canada [3]. The present ML model and tissue testing could be combined to dig further into nutrient factors limiting the yields of vegetable crops.

### 4.5. Potential Economic and Environmental Gains

Searching for the right fertilization rate and timing is challenging in intensive vegetable production where fertilization cost is small compared to crop value. Nevertheless, organic soils show great capacity to supply nitrogen to the crop [21] and thus have high biological buffering capacity [2]. On the other hand, the strong legacy of continuous P overfertilization in organic soils impacts profoundly the P cycle [41].

In the present study, all N/P ratios were less than 10, indicating N-limited biomass production even in high N-supply organic soils [21]. The moderate crop response to added N compared to the absence of crop response to added P in the high soil test P cultivated organic soils indicated fast acquisition of P by the plant from easily available P forms, compared to N acquisition that is mediated by organic matter mineralization. The N/P ratio of vegetable crops grown in organic soils should be interpreted as relative excess rather than relative limitation as currently diagnosed in relation to growth-limiting factors [17].

#### 4.5.1. Nitrogen

Predicted crop response to added N, N offtake, imbalanced N/P ratio, and $N_2O$ emissions are strong arguments to support the sustainability of vegetables grown in organic soils. Net N nitrification rates in fast-mineralizing Quebec organic soils are like those of fast-nitrifying organic soils worldwide [21,33]. High N mineralization rates up to nearly 600 kg N $ha^{-1}$ [21,42] may exceed N offtake by the crops (Table 4). Nonetheless, the high N offtakes (147–282 kg N $ha^{-1}$) may exceed soil N supply for short periods during the growing season, requiring supplemental fertilization to cover crop N requirements.

Fast nitrification rates also lead to high $N_2O$ emissions [43]. Compared to emission rates of 4.0–11.7 kg $N_2O$ $ha^{-1}$ $year^{-1}$ reported in cultivated organic soils of northern Europe [44], the $N_2O$ emission rate in cultivated organic soils of eastern Canada ranged from 3.6 to 40.2 kg $N_2O$ $ha^{-1}$ $year^{-1}$ [42]. Using 273 as conversion factor, this represents one to 11 Mg $CO_2$-equivalent $ha^{-1}$ $year^{-1}$ as $CO_2$-equivalent off-farm impact on climate change.

The $N_2O$ emission rate depends on mineral N content in the soil, gas concentrations, temperature, water table height, soil water content, and irrigation [42]. Fertilization rates up to 150 kg N $ha^{-1}$ may impact $N_2O$ emissions differentially in non-irrigated and irrigated plots, due to plot-specific soil moisture/aeration and carbon and nitrate bioavailability. The N fertilization did not impact $N_2O$ emissions at the irrigated site, likely due to nitrate leaching and limited soil denitrification capacity at high nitrate concentration in the soil [7].

Supplemental N fertilization may be needed because the rate of N mineralization may not be fast enough to meet plant requirements, the $N_2O$ emission rates may be too fast, or the soil microbial biomass may acquire more N to stabilize the N/P ratio constant in high-P soils and maintain homeostasis. As a result, excessive P fertilization could

increase N requirements, combining environmental damages through eutrophication and $N_2O$ emissions.

On the other hand, irreversible carbon losses in organic soils have been compensated in part by biomass additions [45–48] that must also alter the C:N:P stoichiometry in the soil. Soil and fertilizer N and P would thus impact not only the N and P cycles but also trigger soil carbon loss. The C:N:P relationships could be further investigated in organic soils in relation to the N-P fertilization, soil microbial biomass, and carbon loss through decomposition and carbon leaching.

### 4.5.2. Phosphorus

The critical P saturation index has been set at 5% $[P/(Al + 5Fe)]_{Mehlich3}$ molar ratio to inform growers about crop response probability to P additions and the associated risk of surface water eutrophication [10]. Above that critical P index, the P accumulates in organic soils primarily as NaOH-extractable organic and inorganic P forms, $NaHCO_3$-extractable inorganic P forms, and easily leachable resin-extractable inorganic P forms [41], resulting in high risk of P loss to the environment [10]. The P loss between the fall and the following spring was found to average 41 kg $P_{Mehlich3}$ $ha^{-1}$ or 84 kg $P_{oxalate}$ $ha^{-1}$ in the region [49]. The $P_{oxalate}$ form that is close to total P [41] was greater than P offtake for the highest crop yields (Table 4). As a result, P fertilizers can be viewed as a useless source of pollution of surface waters.

Universality tests confirmed the low N/P ratios of the harvested products due to P overfertilization. The P application rates could thus be skipped or reduced substantially with no yield loss. Nonetheless, soil test P should be monitored yearly to check for P saturation index dropping below 5% because crop response to added P has not been documented. Fertilizer trials should be conducted to validate the agronomic significance to P saturation indices below 5%.

## 5. Conclusions

The random forest model was moderately to substantially accurate to predict marketable yields and P offtake of vegetables grown in organic soils, and moderately accurate to predict N offtake and the N/P ratio of the harvested products. The P fertilization in high-P soils proved to be wasteful in independently conducted universality tests. Extra N fertilization that causes nitrate leaching and $N_2O$ emissions can be addressed factually by yield response patterns drawn by the predictive ML model and by universality tests. While the RF model relating target variables to 22 features showed substantial strength, more features may impact the target variables. Universality tests should thus be conducted to verify model's capacity to generalize to local conditions unseen by the model.

Median crop P offtake was similar across crops. Potato showed the highest median N uptake among crops. Median P offtakes were much lower than the reported P loss between the fall and the following spring in the region, indicating large potential for P leaching and surface water eutrophication. The N and P offtakes were impacted by P-related features, indicating an effect of P overfertilization on the N cycle.

The N/P ratio was found to be too low in the harvested products compared to forbs even where N was applied in excess of plant needs. The N/P ratio of vegetables grown on organic soils should thus be interpreted in terms of relative P excess rather than N limitation. This study showed that growers' inclination toward extra N and P dosage to produce vegetables in organic soils can be reduced substantially based on ML model, nutrient offtakes, and crop N/P ratios, contributing to the sustainability of on-farm and off-farm agroecosystems.

**Funding:** The research was funded by provincial programs (Programme de Soutien aux Essais de Fertilisation—PSEF) and Programme de soutien à l'innovation horticole—PSIH) and federal programs (Fonds d'Exploitation des Infrastructures, the Natural Sciences and Engineering Research Council #2254).

**Data Availability Statement:** The database is unavailable due to privacy restrictions.

**Acknowledgments:** The author acknowledges Annie Pellerin for sharing notebooks, and Elizabeth Parent and Catherine Tremblay who built the first version of the database. Thanks are extended to Serge-Étienne Parent, ecological engineer, for advice on ML methods.

**Conflicts of Interest:** The funders had no role in the design of the study; in the collection, analyses, or interpretation of data; in the writing of the manuscript; or in the decision to publish the results.

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
