# Peer review of "Vegetable Response to Added Nitrogen and Phosphorus Using Machine Learning Decryption and the N/P Ratio"

_horticulturae, doi:10.3390/horticulturae10040356_

Round 1

Reviewer 1 Report

Comments and Suggestions for Authors

Congrats to the author on very interesting work!

Author Response

Thank you for positive review. 

The Ms was further improved by adding a graphical abstract, producing clearer figures, correcting typos and improving the flow.

I hope that the revised Ms will be satisfactory.

Reviewer 2 Report

Comments and Suggestions for Authors

Very good paper. Acceptable after very few minor editing revision.

Comments

Title suggestion change: 

Vegetable response to added nitrogen and phosphorus using  modeled through machine learning decryption and N/P ratio of crops

line 165 in Table 4: RMSE rather that RSME

line 210: "there" rather then "yhere"

line 331: delete "likely "  twice

Author Response

Thank you for positive review. 

The Ms was further improved by adding a graphical abstract, producing clearer figures, correcting typos (including yours) and improving the flow.

I hope that the revised Ms will be satisfactory.

Reviewer 3 Report

Comments and Suggestions for Authors

The paper entitled Vegetable response to added nitrogen and phosphorus using machine learning decryption and N/P ratio used random forest model to predict marketable yields, N and P offtakes, and the N/P ratio of vegetable crops, which provides a new insight into scientific fertilization of vegetables. I believe this paper fits the scope of this journal well. The language of the paper is perfect and the context is well-structured. In addition, this paper has detailed data, and the model employed here is robust; I can hardly say no to this paper. All I think that needs to improve is the quality and arrangement of the figures.

Author Response

(The authors gave the same response as above.)
